# Fluid Intake Recommendation Considering the Physiological Adaptations of Adults Over 65 Years: A Critical Review

**DOI:** 10.3390/nu12113383

**Published:** 2020-11-04

**Authors:** Olga Masot, Jèssica Miranda, Ana Lavedán Santamaría, Elena Paraiso Pueyo, Alexandra Pascual, Teresa Botigué

**Affiliations:** 1Department of Nursing and Physiotherapy, University of Lleida, 25198 Lleida, Spain; olga.masot@udl.cat (O.M.); ana.lavedan@udl.cat (A.L.S.); elena.paraiso@udl.cat (E.P.P.); alexandra.pascual@udl.cat (A.P.); teresa.botigue@udl.cat (T.B.); 2Health Care Research Group (GRECS), Biomedical Research Institute of Lleida, 25196 Lleida, Spain; 3Nursing Home and Day Center for the Elderly Balàfia II, Health services management (GSS), 25005 Lleida, Spain

**Keywords:** fluid intake, recommendation, physiological adaptations, older people

## Abstract

The aim of this critical review was to clarify recommended fluid intake for older people. A literature search of published articles and guidelines on fluid intake recommendations until April 2020 was carried out using PUBMED, Scopus, Cochrane, and Google Scholar. In this review, we focused on people over 65 years old at different care levels. The results show that the mean fluid intake ranges between 311 and 2390 mL/day. However, it is difficult to know whether this corresponds to the real pattern of fluid intake, due to the variability of data collection methods. With respect to the recommendations, most international organizations do not take into consideration the physiology of ageing or the health problems associated with an older population. In conclusions, we recommend to follow the guideline of the European Society for Clinical Nutrition and Metabolism (ESPEN) and the European Food Safety Authority (EFSA). ESPEN is the only guideline which takes into account age. It is also based on EFSA recommendations. This authority takes into consideration all fluids consumed (ranging from food to fluids). If it is known that around 20% of all fluids consumed come from food, the result would effectively be that the EFSA recommends the same as the ESPEN guidelines: 1.6 L/day for females and 2.0 L/day for males. The findings could help raise the awareness of professionals in the sector with respect to the required fluid intake of the elderly and, in this way, contribute to avoiding the consequences of dehydration.

## 1. Introduction

Water is the main component of the human body, accounting for over 50% of total body mass [1]. The actual percentage varies, depending on the water losses that take place over the course of the day through respiration, sweat, urine, and feces. A minimum daily fluid intake is, therefore, required to compensate for these losses [2]. Water homeostasis is the key to maintaining a balance between liquid inputs and outputs. It is largely dependent on fluid intake and output and is controlled by the action of baroreceptors and osmoreceptors signaling, via several different pathways [3]. The physiological process found in a healthy adult to compensate for a reduced blood volume is a cascade of reactions known as the renin-angiotensin system. As plasma levels fall, the level of solute concentration in the interstitial fluid increases. Then, due to osmotic pressure, fluid moves outside the cell from the interstitial space. Intracellular dehydration occurs as a result, and then the osmoreceptors command the hypothalamus to manufacture an antidiuretic hormone or arginine vasopressin (ADH) to prevent water loss. Then, it releases it in response to changes in blood volume or serum osmolarity. The main action of ADH in the kidney is to regulate the volume and osmolarity of the urine. ADH secretion is increased during states of increased plasma osmolality [4,5]. More specifically, it acts in the distal convoluted tubule and collecting ducts. In addition to the ADH vasoconstrictive effect, angiotensin II activates the release of the mineralocorticoid hormone aldosterone from the adrenal gland. Aldosterone is a key hormone that is produced by the adrenal gland and enhances Na+ reabsorption through the odium-chloride cotransporter in the distal convoluted tubule and the epithelial Na+ channel. An absence of aldosterone causes urinary Na+ wasting, leading to volume depletion and hypotension. As a result, low blood volume ultimately results in renin release, which has an impact on the renin-angiotensin-aldosterone system. Angiotensin II and aldosterone restore blood volume and pressure by affecting Na+ and water reabsorption and through constriction of blood vessels. The end product of the cascade of the renin-angiotensin system has a direct impact on water retention, with ADH and angiotensin II acting on the kidneys to conserve sodium and to facilitate the osmotic reabsorption of water [6]. However, the evidence shows that older people may exhibit diurnal fluctuations in ADH secretion. This would explain the increased nocturnal diuresis observed in this population [7]. At this point, it can be said that any changes in blood volume are offset physiologically. Water homeostasis, then, acts in parallel to replenish these liquids [8]. When intracellular dehydration occurs, the osmoreceptors also instruct the hypothalamus to induce stimulation of the sensation of thirst [9]. This can also be activated by other conscious non-homeostatic perceptions such as perceived mouth dryness or wetness, stomach emptiness, or distention [10]. Older people have a weaker sensation of thirst and one that is also activated later than in other adults, despite the body’s confirmed need for fluids. Therefore, they require more intense stimuli to make them feel thirsty. A reduction in fluid intake and the sensation of thirst have also been observed in the elderly both at baseline and in response to various different osmotic stimuli. Therefore, it is common to observe a relative hypodipsia in the elderly. For all of these reasons, the osmolar threshold required to perceive a sensation of thirst is greater in the elderly than in other adults.

Due to the pathophysiology of older people and how it affects their water homeostasis, it would seem essential to have fluid intake recommendations that specifically consider the physiological needs of adults over 65 years old. For this reason, the aim of this critical review is to clarify the fluid intake recommended for older persons.

## 2. Materials and Methods

A critical review means thinking carefully and clearly and taking into consideration both the strengths and weaknesses of the material under review. An effective critical review presents, analyzes, and synthesizes material from diverse sources. Its product perhaps most easily identifies it, typically manifesting itself as a hypothesis or a model, rather than an answer [11].

In the present case, the critical review framework adopted was based on the methodological model of Grant and Booth [11]. Relevant studies were identified by searching recent literature, published until April 2020. O.M. and J.M. worked together to search the PUBMED, Scopus, Cochrane, and Google Scholar databases using the following terms: ”dehydration” and ”hydration’; ”hypernatremia”; ”thirst”; ”drink*”; ’fluid intake”; ”low fluid intake”; ”aged”; ”elderly people”; ”older people”; ”hospital*”; ”community care”; ”primary care”; ‘”care home”; ”residential facilities”; ”nursing home”, and ”long-term care facilities”. No limits were put on the language of publication, because few research projects have been carried out in this area. The criteria for the inclusion of papers were the following: the most recently published articles, recommendations and guidelines on fluid intake, and classic texts. However, no exclusion criteria were established.

## 3. Factors Associated with the Water Balance in Older People

The physiological processes described above are conditioned by ageing, as renal function and the ability to feel thirst are altered as a person gets older [12,13]. Kidney function declines with age, and consequently, the ability to concentrate urine and retain fluid decreases. In this respect, Davies et al. [13] compared the response of water balance regulation mechanisms in the rehydration of a group of young adults with that of a group of older people. They found that men aged over 70 had lower resting ADH levels than those under 40. Similarly, dehydration vasopressin levels rose faster in older men (*p* = 0.02). However, it seemed that ADH was activated at similar osmolar values in young and aged adults. Hughes et al. [14] employed linear regression to define the ADH threshold. As a result, the mean ± 95% CI ADH release threshold was very similar to that of thirst in young adults, being 284.3 ± 0.71 mOsm/k, data which was similar to those by Bouby and Fernandes [15] and Miescher and Fortney [16]. Consistent with the latter classic text, authors dissociated thirst ratings from fluid intake in the case of older men previously subjected to prolonged passive heat stress. Older people were noted to be thirstier when their plasma osmolality values were higher. This was greater in older than in younger men (*p* < 0.01), with a mean serum osmolality of around 292 and 287 mosm/kg, respectively. In other words, when both volume and tonicity were affected by dehydration, older people exhibited a higher plasma osmolality (homeostatic controls) [5,8]. Another interesting study on thirst and its activation was carried out by Davies et al. [13]. They found that, although hypertonic saline infusion loading caused significant decline (*p* < 0.001), there was no variation with age. Nevertheless, the perception of thirst during the osmotic loading experiment was perceived differently by the different age groups (*p* < 0.0001). The authors believed that the ambiguous thirst results could have been due to this being a complex and subjective mechanism. One possible explanation could be associated with whether or not the stimulation of thirst was due to the homeostatic water process. In a qualitative study, older people revealed that their experience of drinking was increased by actually drinking, whether or not they needed to; it was important for this to be a pleasurable and social experience [17]. However, it could, therefore, be argued that not all the processes of thirst activation found in the elderly were affected in the same way. Therefore, it is necessary to adopt subtle experimental approaches in order to unravel the intricacies of this approach [18].

The water balance of older people can additionally be compromised by other factors. According to a recent review by our research group [19], and using comprehensive geriatric assessment-based risk factors [20], the sociodemographic factors that were found to be most closely associated with dehydration were age and being female. As for clinical factors, infections, renal and cardiovascular diseases, and end-of-life situations were found to have a strong association. The most associated functional factor was the inability to deal with daily life activities, while the most significant mental health factors were dementia and behavioral disorders. Finally, the social factors associated with dehydration were institutionalization, skilled care level requirements, and wintertime. In short, maintaining an appropriate level of hydration in older people is a constant challenge [1].

## 4. Fluid Intake of the Elderly

It has been shown that older people consume less water than younger people. In a National Center for Health Statistics data brief [21] on the daily water intake of U.S. citizens, the relevant data of 20,293 participants in the National Health and Nutrition Examination Surveys (NHANES) of 2009–2012 were analyzed and statistically compared, showing that fluid intake of those aged 60 and over was lower than that of the other age groups (*p* < 0.01).

Bearing in mind this premise, we additionally considered whether differences existed in the studies at different care levels (Table 1). For example, as most fluid intake studies at hospital level focus on the occurrence or not of dysphagia in post-stroke patients, it is difficult to find age-associated evidence. In a recent study by Buoite Stella et al. [22], an analysis was undertaken of fluid intake by means of a diary in which patients and caregivers recorded all fluid intake during a hospital stay. Fluid intake was considered to be low if below 25 mL/kg/day. The mean oral fluid intake in patients with dysphagia was 511 mL/day (SD 560), and 1780 mL/day (SD 472) in patients with no dysphagia (*p* < 0.01). Differences were observed in the prevalence of inadequate fluid intake (dysphagia 93.8% and no dysphagia 63.9%) (*p* = 0.025).

In reference to living at home, Namasivayam-MacDonald et al. [23] estimated total fluid intake over three non-consecutive days (24 h) including one weekend day. The amounts of liquid consumed were recorded by caregivers during the day and by family members at night, with inadequate intake considered to be <1500 mL/day. The mean daily fluid intake ranged from 311 to 2390 mL/day (mean 1104.1, SD 379.3), with 88% of cases recorded as being inadequate. Moreover, patients aged 85 or older were found to be eight times more likely to have a low water intake (OR = 7.84, *p* < 0.05). Contrasting results were found in a study by Lindeman et al. [24]. They asked the study participants “How many glasses of liquids (including water, juice, coffee, tea, milk, wine, beer) do you drink per day?” Possible answers were (a) less than three glasses per day, (b) three to five glasses per day, or (c) six or more glasses per day. The corresponding results to the options (a), (b), and (c) were 2.5%, 26.0%, and 71.5%, respectively. Therefore, one-third of the participants had low fluid intake (the adequate fluid intake, according to the authors, was (c)). In another study by Picetti et al. [25], water intake was also measured by means of a question to the participants, on this occasion “How much liquid do you consume each day?” with possible answers of (a) one to three glasses, (b) four to six glasses, (c) seven to nine glasses, or (d) more than nine glasses. In this case, optimal fluid intake was considered to be >six glasses per day (options (c) and (d)). In total, 56% of the participants reported consuming >six glasses of fluid/day, whereas 9% reported drinking ≤ three glasses, and 35% four to six glasses.

Another interesting study was conducted in a care home setting [26]. Fluid intake records were recorded by the participants themselves in a 24 h drinks diary and by care staff using the care home’s usual fluid intake chart. These were, then, compared with 24 h drinks intake as assessed by the researcher’s direct observation (reference method) during waking hours (06:00–22:00) and through self-reporting with care staff verification for the remaining hours. The mean fluid intake was 1989.0 (SD 757.8) and, based on the records in the drinks diary, 13.64% of the participants had low water intake according to the recommendations of the European Food Safety Authority (EFSA) [27]. The self-completed drinks diary record showed a strong correlation with researcher direct observation (Pearson correlation coefficient r = 0.93, *p* < 0.001, mean difference 163 mL/day). 

Finally, at a long-term residential care level, numerous studies have shown that the daily fluid intake of older people in residential care homes is below the recommended daily requirement [28,29,30]. In a recent study by Botigué et al. [19], although an average daily fluid intake was reported of 1768.5 mL/day (SD 542.2), it was found that 34% of residents drank <1.5 L/day. These results were considerably lower than those reported in other studies [18,20], which could be due to the way the data were collected. In a study by Botigué et al. [29], fluid intake data were collected over a week and for 24 h a day, whereas in Reed et al. [30], data were collected only at mealtimes and low fluid intake was considered to be ≤8 oz in a single meal, with 61.8% found to have inadequate fluid intake. However, Jimoh et al. [28] collected data, during one 24 h period, through direct observation during the day and staff reports during the night, and followed the EFSA criteria to define correct fluid intake [27]. They found mean total drinks intake was 1787 mL/day (SD 693) and that 45% did not achieve the EFSA fluid intake goals.

Given the data presented above, there are clearly many differences in fluid intake records and data collection methods, and therefore it is difficult to determine the actual water intake profile of older people. In addition, it is known that there are also other ways to record water balance, such as measuring urine output, which have not been reflected in these studies [31]. However, knowing that such methods may not be appropriate for this population due to the health conditions that often accompany aging (e.g., urinary incontinence and dementia) [32], it is true that the evidence suggests that 24 h urine osmolality is a good indicator of proper hydration status in adults. According to Perrier et al. [33], a 24 h urine osmolality ≤500 mOsm/kg may be a simple indicator of optimal hydration, representing a total daily fluid intake adequate to compensate for daily losses, ensure urinary output sufficient to reduce the risk of urolithiasis and renal function decline, and avoid elevated plasma vasopressin concentrations mediating the increased antidiuretic effort. Furthermore, there are other methods that are more appropriate for detecting alterations in hydration status. Various analytical tests, signs, and symptoms have been considered that can help to detect dehydration in the elderly [34]. Since there is no gold standard [35], it is necessary to clarify which indicators of dehydration should be included in this group, such as blood and urinary tests to evaluate dehydration in older people. Regarding blood tests, dehydration can be detected through the analysis of Na+ serum, the blood nitrogen/creatinine ratio (BUN/Cr), serum osmolarity, and urine tests. In older people, hyponatremia is a common electrolyte disorder; it is defined as a serum Na+ level of <135 mmol/L [36,37]. Although no single cut-off point for hypernatremia has been defined for the elderly, in a review of the literature, Shah et al. [38] concluded that the range for hypernatremia could range from 140 to 150 mmol/L. With reference to BUN/Cr; this is an indicator of dehydration when there is an increase in BUN, but Cr is normal (> 15:1). It should, however, be stated that other authors have used other cut-off points. For instance, Wu et al. [39] and Bennett et al. [40] used the value ≥ 20:1; Culp et al. [41] used ≥ 21:1, and Mentes [42] used ≥ 25:1. The most commonly used blood test is serum osmolarity. As explained in the Introduction section, serum osmolarity is a key element in the water homeostasis balance. There are many equations with which to calculate this. In fact, some studies have identified up to 35 different formulas [43,44]. The one that is considered to be the best for the elderly is that developed by Khajuria and Krahn [45], because it is able to predict measured serum osmolality in frail older people both with and without diabetes, poor renal function, dehydration, and impaired health, as well as cognitive and functional status [43,46]. The most widely accepted cut-off values for older people take normal values as 275 to ≤295 mmoL/L, while 295 to 300 mmol/L are indicative of impending dehydration, and >300 mmoL/L is recognized as indicating current dehydration [43,44,46].

Urinary tests can also be useful. When a person does not ingest enough fluids, there is an increase in the specific severity of their urine, which causes it to darken. The color of urine tends to react almost immediately to small changes in the hydration state [47]. Hence, the table produced by Armstrong et al. [48,49]. In a study by Wakefield et al. [50], increasing scores on the urine color chart were moderately and positively correlated with specific urine severity and osmolality. In contrast, other researchers [51] have noted that the specific gravity, color, and osmolality of urine have been widely advocated for screening of dehydration in older adults and that, as a result, these measures should not be used to indicate hydration status in older people (either alone or as part of a wider tranche of tests). In a recent study, Armstrong et al. [52] concluded that low volumes differed from high volumes in terms of urinary biomarkers (e.g., reduced urine volume and increased osmolality or specific gravity) and that a self-assessment of urine color provided useful feedback regarding excessive drinking.

In addition, the threshold below which fluid intake is considered to be low also differs between studies. In some studies, fluid intake is measured by the number of glasses drunk [24,25,30]. If these amounts are extrapolated to L/day, low fluid intake is considered to be below 1.5 L/day, values also used in other studies [23,29]. Some studies have relied on individualized formulas of what an individual should drink [22], while yet others are based on the recommendations of international organizations [26,28].

Referring to this variability, this could have also been due to variations in the length of the data collection period. To establish the real pattern of fluid intake, the most appropriate way of collecting information on fluid intake would have been over 24 h and every day for a week. The importance of this lies in the fact that there are various organizational factors that could influence the final result. One of these would be shift work of the hospitals or nursing homes [53]. Other factors could be informal interactions [53] and insufficient staff ratios [54], which tend to be more frequent on weekends. In addition, if older people lives in their home, their intake is affected by the training of their caregivers, the workload of the family, and the risk of the caregiver claudication. Therefore, it could be said that it is not so much whether the information is collected by measuring glasses or recording the mLs drunk, but whether this record really reflects the real pattern of fluid intake over time.

In view of the variability in considering low fluid intake, it is useful to know the guidelines published by internationally renowned organizations, as well as the standards in place concerning the recommended fluid intake of older people.

## 5. Recommended Fluid Intake for Older People

With respect to the recommendations of international organizations, the World Health Organization [55] recommends 3.7 L/day for males and 2.7 L/day for females, with those over 70 years included in the recommendations for healthy adults over 19 years old. This advice is also supported by the U.S. National Academy of Medicine [56] and the U.S. National Center for Health Statistics [21].

At the European level, the EFSA [27] recommends 2.5 L/day and 2.0 L/day for adult males and females, respectively. However, the European Society for Clinical Nutrition and Metabolism (ESPEN) [57] is the only body which distinguishes between adults and older people. Their recommendation is that older males drink a minimum of 2.0 L/day and females 1.6 L/day. 

Consequently, as most organizations focus on the general population [58], their recommendations may not be appropriate for older people, especially given the physiological characteristics and health issues associated with those of more advanced age. However, other authors opt for the use of individualized recommendation standards, where intake is calculated according to individual needs, trying to adapt the amount recommended as much as possible. 

In this respect, an analysis was undertaken in a review by Vivanti [59] of 11 different formulas or equations for the estimation of water requirements based on a relationship between mL and actual body weight. However, it was not specified whether all of them were applicable to older people. Among these formulas, Gaspar [60] highlighted three for this population as follows: firstly, 30 mL/kg/day [61]; secondly, 1 mL fluid per kilocalorie energy consumed [62]; and thirdly, the sum of 100 mL/kg for the first 10 kg of weight, 50 mL/kg for the next 10 kg, and 15 mL/kg for the remaining kg (Skipper standard [63]).

A closer analysis of these standards reveals certain limitations. The first example would not be useful for thin or obese individuals [64]. Low amounts could be recommended for thin people, even below one liter per day, and large amounts for obese individuals that would be difficult to consume. The problem with the second standard is that it is based on energy metabolism, making it potentially unrealistic because of the complexity of the physiological mechanisms of older people. Finally, the problem with the Skipper standard [63], despite its being considered to be the most effective for older adults, as the consumption of at least 1.5 L/day is recommended irrespective of whether the individual has a low weight, is that, as with the first standard, it does not take into account the large amounts of fluid that overweight people would consume [65].

These three standards were also compared in a study by Kayser-Jones et al. [66]. They concluded that the mean fluid intake of the participating residents of two nursing homes was inadequate due to several factors detected through a descriptive and anthropological study. By way of example, they reported on the behavior of a stroke-affected resident who usually ate her meals in bed in a semi-reclined position. She would often spill her drink and would sometimes dip her fingers into the drink and lick them instead of directly consuming the juice. 

Consequently, individualized fluid intake recommendations do not guarantee that all clinical and environmental factors are taken into consideration. In addition, in the case of overweight individuals, the amount calculated could be excessive. For all these, it is important to know the benefits of being well hydrated.

## 6. Health Benefits of Hydration

Having a proper fluid intake prevents older people from suffering acute health problems. These can include falls [67], fractures [68], pressure ulcers [29,68], constipation [67], urinary infections [68,69,70], and other kidney problems such as stones [40,71], functional impairment, and being undernourished [29]. Moreover, older people who suffer dehydration also risk suffering acute coronary events (1.6% vs. 0.7%, OR 1.16, 95% CI 1.03–1.32), pneumonia (3.4% vs. 1.5%, OR 1.23, 95% CI 1.13–1.34), and thromboembolism (1.8% vs. 0.9%, OR 1.28, 95% CI 1.14–1.42) [72]. As a result, dehydration is associated with an increased risk of suffering disability within four years (OR 2.1, 95% CI 1.2–3.6) [73]. With regard to mental health, a correct level of hydration is important for maintaining cognitive abilities and having a healthier mind which is required to prevent states of confusion or delirium [29,67,70]. An adequate fluid intake is also essential for maintaining chronic medical conditions. Without it, they may become exacerbated [74,75], resulting in conditions such as imbalance in diabetes [76], heart disease [77], drug toxicity [78], kidney stones, or renal failure [40,71]. 

Such physiological imbalances also increase the risk of repeated hospitalizations [79]. This explains why dehydration is listed as one of the 20 most common diagnoses reported by the U.S. Agency for Healthcare Research and Quality [80]. This agency has reported more than 300,000 hospital admissions for dehydration involving older people. 

Related to mortality, Warren et al. [74] found that approximately 50% of older people hospitalized for acute and chronic dehydration died within one year of admission. Patient mortality may also be as much as seven times higher than for those who are not dehydrated [75,81]. 

In view of all of this, low fluid intake in older people is extremely costly. Focusing on hospitalization, a recent review of the associated economic burden [82] indicated that dehydration can increase patient care costs from 7% to 8.5%, especially among those with moderate to severe hyponatremia. Moreover, a recent study conducted in the USA [83] concluded that the average total hospital charge was USD 7442 for hospitalized older patients with a principal diagnosis of dehydration. 

In conclusion, having an appropriate fluid intake provides mental, physical, and general health benefits, and ensures a higher quality of life in older people [84]. It can also imply great money savings for healthcare systems. However, it must be noted that all these health benefits of hydration can be compromised by overhydration.

## 7. Overhydration: A Potential Problem

Overhydration or “excess total body fluid” is a pathogenic condition. It occurs when the body takes in or holds onto more fluid than the kidneys can turn into urine. In older people, a saturation point can be reached if more than eight glasses of water (2.0 L) are drunk per day and, according to various authors, there is no evidence that drinking larger amounts has any health benefits [24,85]. Physiologically, the kidneys of older people do not have the same filtering capacity as those of a healthy younger adult. Acute water intoxication toxicity can take place when large amounts of liquid are rapidly consumed, and the maximum kidney excretion rate is exceeded. For older people, this rate ranges between approximately 0.7 and 1.0 L/h. In this respect, Hemmelgarn et al. [86], in a cohort study that included 10,184 people >66 years of age with a minimum two-year follow-up, concluded that kidney filtering capacity declined on average each year by 0.8 mL/min/1.73 m^2^ in females (95% CI 0.6–1.0) and 1.4 mL/min/1.72 m^2^ in males (95% CI 1.2–1.6). 

With respect to the factors associated with overhydration, a study by Picetti et al. [25], in which 170 individuals over 60 years of age who lived in the community were surveyed about their hydration awareness, found that approximately one-third were unaware that fluid overload occurs in heart failure (35%) or kidney failure (32%) (*p* < 0.05), among others. In this respect, the EFSA [27] considers that fluid intake should be limited in these clinical situations, though it does not specify when. 

With respect to heart-related pathologies, the U.S. and European cardiology societies attach great importance to this question [87,88], stipulating a limit in patients with moderate or severe heart failure symptoms of 1.5–2.0 L/day [87], and ensuring that a patient with chronic heart failure and taking diuretics is not dehydrated [88].

However, a meta-analysis carried out by De Vecchis et al. [89] found that liberal fluid consumption (>2.0 L/day) did not appear to exert an unfavorable impact on adult patients with chronic heart failure (mean age 69.5 years, age range 60–75). Patients whose fluid intake was limited had similar rehospitalization and mortality rates to those with no fluid intake limit. 

As for kidney pathologies, it is also unknown whether a higher than usual water intake has an effect on older patients with chronic kidney disease [90]. The results of the clinical trial, which was conducted on adults (mean age 65.0 years, SD 11.8) with stage 3 chronic kidney disease [91], showed that coaching patients to increase fluid intake by 1.0 to 1.5 L/day did not significantly slow the decline in kidney function after one year as compared with a control group who continued with their usual fluid intake. 

In view of these results, liberal fluid intake may not have an unfavorable impact on these pathologies where water intake restrictions are typically applied. 

## 8. Conclusions and Implications

Given all of the above, it is clear that much is unknown about the optimal recommendation for fluid intake of older people. Some of the guidelines given by international organizations do not take into account the physiological characteristics of older people. This is also true of the standards which have been published. Their calculations are mostly weight based and may result in the recommendation of too low amounts of fluid or high amounts which may be impractical. In addition, little research has been undertaken focusing on low fluid intake and its effects on older people. Finally, it is difficult to be certain as to whether the reported fluid intake profile corresponds to reality, as records are rarely taken for a whole week, are sometimes collected randomly, are not collected for 24 consecutive hours, or are simply based on participants’ memory of what they drank and when it was consumed. 

Additionally, according to the EFSA [27], adequate fluid intake for older people should not be solely based on observed intake, but should also take into account water from ingested food. Moreover, the American Medical Directors Association [92] recommends that certain risk factors associated with low fluid intake need to be considered to ensure an adequate fluid intake. These include decreased cognitive function, kidney conditions, and oral intake restrictions, all of which are highly prevalent among older people. There is growing evidence that people with a health problem who drink less than their daily requirements will find their condition worsening, especially those who drink <1.0 L/day [29].

With respect to the standards that have been published, it has been seen that they do not take into account the particular characteristics of older people and may not be a reliable indicator. A number of metabolic changes take place in older people, particularly in the female population, which tend to reduce muscle mass and increase lean mass, thereby triggering a decrease in overall body water content [93]. However, this group is already based on a smaller proportion of water than men. This explains why the fluid intake recommendations for the female group are lower than for the male group. The standards may be useful when personalized fluid intake recommendations are required, as could be the case for patients with kidney or heart diseases. According to the American College of Cardiology Foundation and the American Heart Association [88], continuous checks on diuretic dosage could be key to applying the personalized benefits offered by these standards, and therefore mitigating the possible negative effects of excess fluid intake above 2 L/day. As well as such checks, the monitoring of weight changes, physical/biochemical parameters, and urine volume outputs should also be performed. 

Therefore, while there is some uncertainty as to an appropriate recommendation for fluid intake for older people, this critical review clearly provides evidence that most international organizations do not take into account the physiology of ageing in their recommendations, nor the health problems that typically affect older people. However, in view of the all the evidence that has been analyzed, it may be concluded that older people should drink between 1.5 and 2.0 L/day [57,58,87,88]. More specifically, it is recommended to follow the ESPEN [57] and EFSA [27] guidelines. ESPEN [57] is the only guideline which takes into account age and enjoys a 96% consensus among experts. It is also based on EFSA [27] recommendations. This authority [27] takes into consideration all fluids consumed (ranging from food to fluids). If it is known that around 20% of all fluids consumed come from food [94], the result would effectively be that the EFSA [27] recommends the same as the ESPEN guidelines [57], i.e., 1.6 L/day for females and 2.0 L/day for males. In this way, care is taken to avoid the negative effects of dehydration while, at the same time, ensuring that people with heart and kidney diseases are equally safe with both recommendations. Nonetheless, these recommendations do not preclude the possibility of further research to verify their effectiveness and safety in terms of the health of older people. The main goal for the future should be to individualize treatments according to a patient’s physiology and the health condition of a patient. Furthermore, it should not be forgotten that the pattern of fluid intake observed in Table 1 differs according to the level of care that an older person receives and possibly due to environmental factors, staff care, and health status of each person. Such research studies may be experimental in nature, comparing a control group with usual fluid intake and a second group with fluid intake proposed by the study designers, or longitudinal in nature, observing different fluid intakes and their effect over time on the health of the participants. In both cases, clinical variables such as urinary tract infections, pneumonia, pressure ulcers, hypotension, disorientation, confusion, and electrolyte imbalances (hypernatremia, hyponatremia, and hyperkalemia) need to be monitored to verify the results. In this way, it should also be possible to ascertain whether the recommendations made for the elderly, apart from varying between adults and sexes, take into account the place where this person lives. In addition, in this type of study, it should also be possible to identify the effects of the baseline health condition of the participants and the physiological ageing of the human body.

## Figures and Tables

**Table 1 nutrients-12-03383-t001:** Characteristics of the studies that analyzed the fluid intake of older people.

Authors (Year)	Methodology	Study Location	Sample Size and Characteristics	Assessment Tool	Guidelines Meet for Considering Low Fluid Intake	Fluid Intake Volume
Buoite Stella et al. (2019) [22]	Retrospective observational study	Italy	*n* = 95 hospitalized Stroke patients admitted to a hospital stroke unit	Diary in which patient and caregiver recorded all fluid intake during the hospital stay.	25 mL/kg/day	Mean oral fluid intake: - with dysphagia: 511 mL/day (SD 560) - without dysphagia: 1780 mL/day (SD 472) - (*p* < 0.01)
Namasivayam-MacDonald et al. (2018) [23]	Cross-sectional study	Canada	*n* = 622 living at home	Liquid consumption was recorded by caregivers during the day and by family members at night. It was recorded in three non-consecutive days including one weekend day over 24 h.	<1500 mL/day	- 311–2390 mL/day (mean: 1104.1, SD 379.3) - 88% of intakes were inadequate
Lindeman et al. (2000) [24]	Cross-sectional study	USA	*n* = 883 living at home	Participants were asked ”How many glasses of liquids (including water, juice, coffee, tea, milk, wine, beer) do you drink per day?” Answers were (a) less than three glasses per day, (b) three to five glasses per day, (c) six or more glasses per day.	(a) and (b)	1/3 of the participants had low fluid intake.
Picetti et al. (2017) [25]	Cross-sectional study	USA	*n* = 170 living at home	Participants were asked ”How much liquid do you consume each day?” Answers were (a) 1–3 glasses, (b) 4–6 glasses, (c) 7–9 glasses, (d) more than 9 glasses.	≤6 glasses per day ((a) and (b))	44% reported consuming ≤6 glasses of fluid/day
Jimoh et al. (2015) [26]	Cross-sectional study	UK	*n* = 170 in a care home	Three parallel records were compared: Participants; Staff; RDO.	EFSA recommendation	RDO data: - 1989.0 (SD 757.8) - 13.64% of the participants had low water intake
Botigué et al. (2019) [29]	Cross-sectional study	Spain	*n* = 53 institutionalized in a nursing home	Staff collected for 24 h per day over a period of 1 week.	<1500 mL/day	- 1768.5 mL/day (SD 542.2) - 34% drank less than <1500 mL/day
Jimoh et al. (2019) [28]	Cross-sectional study	UK	*n* = 22 institutionalized in nursing homes	Data were collected during one 24 h period, through direct observation. During the day, it was observed by researchers and staff reports during the night.	EFSA recommendation	- 1787 mL/day (SD 693) - 45% did not achieve the EFSA fluid intake goals
Reed et al. (2005) [30]	Cross-sectional study	USA	*n* = 407 institutionalized in nursing homes	Data were collected only at mealtimes.	≤8 oz in a single meal	61.8% found to have inadequate fluid intake.

N, participating population number; RDO, researcher direct observation; EFSA, European Food Safety Authority, SD, standard deviation.

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
