# Peer review of "Fluid Intake Recommendation Considering the Physiological Adaptations of Adults Over 65 Years: A Critical Review"

_nutrients, 2020, doi:10.3390/nu12113383_

Round 1

Reviewer 1 Report

This is an interesting and important topic, but authors need to reconsider the purpose and the structure of the review.

Section 1.  The opening section on Water Balance Physiology is superficial and needs to be much more precise (if renal physiology is to be included).

Line 39 – The term ‘ordering’ is inappropriate

Line 41 – State the action of ADH on the kidney

Diurnal fluctuations in ADH should be considered.

The 2nd section is disjointed.  The 1st paragraph of this section refers to a single experiment that is poorly explained (Line 53 Not clear what is meant by lower resting levels of ADH).  The next paragraph reports sociodemographic factors, again this is too superficial.

The 3rd section is headed ‘Results’, yet no methodology has been presented to this point, so results of what? The rationale, the search strategy and analysis approach used needs to be described.   The results section describes 8 studies, is this the totality of evidence? 

Section 4 reports recommended fluid intake, but intuitively this should come earlier in the introduction.

The conclusion is that the ESPEN guidelines should be followed, however there is no justification for this and the only mention of ESPEN in main part of the review is that it ‘is the only body which distinguishes between adults and the older people’. This single statement tells the reader little and is insufficient basis to reach the conclusion made. 

Reviewer 2 Report

In this manuscript, authors have aimed to clarify the fluid intake recommendations for the 65 years of age and older population.  They have argued that due to physiological aging and health complications that they should have a separate recommendation than other healthy adults.  To make the argument clearer the following are suggested:

  1. Authors reported that age and being female are associated with dehyration. Please further clarify what the rationale from the association?  If being female is associated with dehydration, why would the current recommendations  suggest a significantly lower amount for females?
  2. It is unclear how many current recommendations have been found in their search. Please provide a summary of the search including the number of studies, types of studies etc. A summary table may be helpful here.
  3. It is perhaps helpful to include the "individualised recommendations" to compare to those "general population" recommendations. This will clarify and enable comparison between the recommendation.
  4. Assessing fluid intake is complicated as pointed out by the authors.  The manuscript covers the issues in self-reported and carer-reported types of assessment methods, but have not discussed other methods such as measured urine output.  Please include in the discussion.
  5. Lastly, it is important to emphasise that the recommendations should be based on physiology and health conditions and not on the general intake from observational studies. This is a complex issue even in healthy young adults.  I would also emphasise that the living environment for the elderly will also be very important (e.g. community dwelling vs. nursing home vs. hospitalisation). Perhaps guidelines for each setting and health conditions will be required. 

Reviewer 3 Report

Thank you for critical review. This paper identifies the lack of robust evaluation and recommendation for hydration state for older adults. I think the exploration of older adults through care setting is a useful assessment. 

For edits please consider including a short section in the main body of the paper referring to your literature search methods, what resources were used (you outline these in the abstract but not the text body) what inclusion and exclusion criteria were used for identified results, who conducted the result appraisal, what factors were used to determine inclusion in the final review. 

At the start of section 3 -Results please identify how many relevant papers were included and extracted, and what grouping was determined to discuss in the paper i.e. care setting. 
